# VOGDB—Database of Virus Orthologous Groups

**DOI:** 10.3390/v16081191

**Published:** 2024-07-25

**Authors:** Lovro Trgovec-Greif, Hans-Jörg Hellinger, Jean Mainguy, Alexander Pfundner, Dmitrij Frishman, Michael Kiening, Nicole Suzanne Webster, Patrick William Laffy, Michael Feichtinger, Thomas Rattei

**Affiliations:** 1Centre for Microbiology and Environmental Systems Science, University of Vienna, 1030 Vienna, Austria; 2Doctoral School of Microbiology and Environmental Systems Science, University of Vienna, 1030 Vienna, Austria; 3Armaments and Defence Technology Agency, Austria; 4Genoscope, 91000 Evry Cedex, France; 5Department of Bioinformatics, School of Life Sciences, Technical University Munich, 85350 Freising, Germany; 6Australian Institute of Marine Science, PMB no3 Townsville MC, Townsville 4810, Australia; 7Institute for Marine and Antarctic Studies, University of Tasmania, Hobart 7000, Australia; 8Australian Centre for Ecogenomics, University of Queensland, Brisbane 4072, Australia

**Keywords:** virus genomes, protein families, comparative Genomics, orthologous groups, genome annotation, genome analysis

## Abstract

Computational models of homologous protein groups are essential in sequence bioinformatics. Due to the diversity and rapid evolution of viruses, the grouping of protein sequences from virus genomes is particularly challenging. The low sequence similarities of homologous genes in viruses require specific approaches for sequence- and structure-based clustering. Furthermore, the annotation of virus genomes in public databases is not as consistent and up to date as for many cellular genomes. To tackle these problems, we have developed VOGDB, which is a database of virus orthologous groups. VOGDB is a multi-layer database that progressively groups viral genes into groups connected by increasingly remote similarity. The first layer is based on pair-wise sequence similarities, the second layer is based on the sequence profile alignments, and the third layer uses predicted protein structures to find the most remote similarity. VOGDB groups allow for more sensitive homology searches of novel genes and increase the chance of predicting annotations or inferring phylogeny. VOGD B uses all virus genomes from RefSeq and partially reannotates them. VOGDB is updated with every RefSeq release. The unique feature of VOGDB is the inclusion of both prokaryotic and eukaryotic viruses in the same clustering process, which makes it possible to explore old evolutionary relationships of the two groups. VOGDB is freely available at vogdb.org under the CC BY 4.0 license.

## 1. Introduction

Viruses are a diverse group of biological entities that share the property of being obligate cellular parasites. Unlike in cellular organisms, no common genes or gene families are shared between all viruses [1]. This raises fundamental questions about virus ancestry and evolution. Moreover, the number of viruses on earth is huge (more than $10^{31}$ particles) [2,3], and it is estimated they carry between $10^{8}$ and $10^{10}$ unique genes [4]. Most of the viral diversity is currently unexplored, and for the most sequenced viral genes, little is known about their function [5].

Viral genes not only encode a high number of different functions, which leads to a huge diversity of viral genomes, but also form heterogeneous groups of genes having similar function [6]. Due to the nature of viral lifestyle and their quick replication, mutations and selection, viruses explore the sequence space of genes in less evolutionary time than cellular organisms do [7,8]. Because of the heterogeneity of viral proteins, it is often difficult to find homologs in databases by traditional bioinformatics, such as pair-wise sequence alignments.

The computational inference of gene homology is valuable for annotating genes that are known from their sequence, but have not been experimentally characterized. Homologous genes have diverged from a common ancestral gene and are likely to have same or similar functions in different organisms. A particularly informative computational observation is gene orthology. Orthologous genes have diverged from a common ancestor by a process of speciation (as opposed to the gene duplication in paralogy). Orthologous genes are more likely to keep the ancestral function [9]. Orthologous genes from multiple organisms form orthologous groups. Homologous relationships are deduced from sequence comparisons due to the assumption that important sequence motifs will stay conserved during evolution [10]. However, due to the absence of universal phylogenetic markers for all viruses and frequent horizontal gene transfers between viruses and viruses as well as viruses and hosts, no universal concepts for the orthology of viral protein families are so far available in bioinformatics.

Due to quick viral evolution, it is often impossible to detect homology by the pair-wise alignment of two protein sequences, especially for proteins that diverged longer ago. However, by building a sequence model based on the group of easily detectable homologs, a conserved pattern becomes discernible, which can be used to connect more distant groups [11]. This approach is widely used by databases that cluster together viral proteins, including pVOG [12], which focuses on prokaryotic viruses, as well as the viral sequences of eggNOG [13]. The PHROGs database [14] clusters phage genomes in two steps, first by grouping them based on the direct sequence comparison and later by clustering group Hidden Markov Models (HMMs) to capture remote homology. However, none of these databases represents the high number and broad diversity of virus genome sequences available to date.

We therefore introduce VOGDB, which is a comprehensive database of virus orthologous groups, virus protein families and virus protein structural folds. VOGDB provides these three layers of homologous groups for all viral proteins from RefSeq genomes [15]. The layers are intended to gather proteins with the increasing evolutionary distance reflected in the higher sequence divergence. Contrary to the prokaryotic genomes from RefSeq, where PGAP [16] is used for the submission and consistent reannotation of genomes, virus genomes in RefSeq may keep their annotation from their GenBank [17] submission. VOGDB, making use of all virus genomes from RefSeq, addresses the potential problem of inconsistent and outdated annotation by filtering and partial reannotation in order to ensure a higher quality of final clusters.

## 2. Materials and Methods

### 2.1. General Concept

The first layer of the VOGDB is constructed by all-against-all pair-wise sequence comparisons and represents the easily detectable homologs. The second layer is created by clustering sequence models (HMMs) from the first layer to capture the homology of proteins that diverged beyond the point where homology can be detected by pair-wise alignments. In the third layer, we group together families from the second layer by their shared features within predicted 3D structures. This layer represents remotely homologous groups whose members diverged to a degree that sequence comparison methods cannot detect their similarity anymore. As there is no standard way to validate viral orthologous groups, we suggest an approach based on the homogeneity of functional and structural annotations in terms of SwissProt [18] keywords and SCOPe [19] superfamily labels. The calculation of homogeneity was also applied to other similar databases (pVOG, PHROGs and COG) to compare if VOGDB shows similar homogeneity despite its higher number and wider diversity of genome sequences. pVOG and PHROGs are databases with viral proteins and are directly comparable to the first and second layers from VOGDB. The COG database contains prokaryotic proteins grouped by orthology and was included as a control.

### 2.2. Preprocessing of Input Data

#### 2.2.1. Input from RefSeq

The input data are all of the complete viral genomes from RefSeq [15], which have at least one protein annotated. Around 98% of records from RefSeq enter the VOGDB pipeline, meaning VOGDB represents almost the entire viral portion of RefSeq. All sequence records with the same taxonomy ID, strain and isolate are considered one genome in VOGDB, which are further called VOGDB genomes.

#### 2.2.2. Polyproteins

Polyproteins are present in DNA viruses and almost all RNA and retroviruses. A polyprotein is translated as a large polypeptide from a single ORF and is later cleaved into functional proteins [20]. At the moment, no general computational strategy exists that would predict the cleavage sites in polyproteins and find the borders of the individual peptides. The iterative approach LAMPA annotates multidomain proteins and addresses the problem that statistical significance is related to the length of domains [21]. We have developed a strategy to annotate the individual peptides from the polyprotein sequence without prior knowledge of conserved domains. First, individual peptides originating from a polyprotein or from RefSeq records that have been validated by the VOGDB team are collected in a peptide reference database. Second, non-annotated or incompletely annotated polyproteins are then reannotated by the best non-overlapping pair-wise sequence alignments against the peptide reference database. Within VOGDB, annotated or reannotated peptides replace the respective segments of their initial polyprotein records and together with the rest of the proteins are called VOGDB proteins.

### 2.3. Creation of the First-Layer Clusters—VOGs

VOGDB proteins are used as the input to the COGSoft pipeline with the aim of constructing clusters of recently diverged proteins [22]. In short, an all-against-all PSI-BLAST [23] search is conducted followed by the COGtriangles [22] procedure to find orthologous groups. We use the strict clustering, which does not allow for a single protein to be a member of multiple clusters. For each orthologous group, a multiple sequence alignment of all member proteins is calculated using Clustal Omega [24]. Scores according to the minimum reporting standard for multiple sequence alignments are obtained using the program alistat [25]. From the multiple alignment, we calculate Hidden Markov Models (HMMs) using hmmbuild from HMMER [26]. The resulting groups are called VOGs to reflect that they are a viral equivalent to orthologous groups.

#### 2.3.1. Functional Annotation

Annotations of VOGDB clusters are made with the aim of describing most of the cluster members as specifically as possible, and therefore, we are using a consensus of the annotations of the individual proteins as the cluster annotation. During the annotation procedure, we prefer manually curated functional information over computationally inferred annotation. VOGs are functionally annotated, if possible, by deriving functional annotations from hits to the most recent SwissProt [18] database or from the annotations as provided by RefSeq. To retrieve the annotation from SwissProt, we used BLAST [27] to search the SwissProt database with the members of a VOG. For an individual protein from a VOG, we retained the functional annotation of a maximum of 5 hits if the e-value was less than $10^{-10}$ and the alignment coverage was more than 90%. All annotations of all proteins in a VOG are collected, and the most common annotation string found for a VOG is used as the annotation for that VOG. In cases when it is not possible to obtain the annotation from SwissProt, we collect annotations of proteins in a VOG as they are in RefSeq and use the most common annotation string as the annotation for the VOG.

As an additional step in the annotation process, we maintain a list of SwissProt keywords with which we associate a functional category. Every functional annotation of VOGs belongs to one or more functional categories: virus replication (Xr), virus structure (Xs), viral protein beneficial for the host (Xh), viral protein beneficial for the virus (Xp) and unknown function (Xu).

#### 2.3.2. Naming

VOG are named with a prefix “VOG” and a number padded with zeroes. To facilitate the comparison of the results between releases, we implemented a stable numbering scheme. VOGs from the older release are compared to the VOGs from the newer release, and the newer VOG receives the name of the largest older VOG for which 50% or more of the proteins are found in the new VOG. For VOGs that do not receive the name from the previous release, a new number is created.

### 2.4. Creation of the Second Layer Clusters—VFAMs

#### Clustering Using MCL

To create the second-layer clusters (VFAMs), we first need to align HMMs of VOGs. The alignment is achieved using the hhalign function from HH-Suite [28]. The HMM–HMM alignments are filtered by three different criteria: maximal evalue of 1 ×$10^{-5}$, minimal HMM probability value of 85 and minimal coverage for both HMMs of 0.7. The scores of alignments that pass all three criteria are used as input to the MCL clustering algorithm [29] where VOGs are clustered with the inflation value of 2. Clustered sequences are aligned with Clustal Omega [24], assessed with alistat [25], and an HMM of the alignment is calculated by the function hmmbuild from HMMER [26]. The functional annotation of VFAMs are obtained in the same way as for VOGs. Naming works the same as for VOGs but with a different prefix: “VFAM”.

### 2.5. Creation of the Third-Layer Clusters—VFOLDs

The third layer of the VOGDB consists of VFOLDs, which are clusters of VFAMs grouped based on the shared structural features. A few experimentally resolved structures of viral proteins are available in the public databases like pdb [30]. Therefore, we used alphafold 2 [31] to predict structures of viral proteins in VFAMs. Since there are more than 500,000 proteins in VFAMs, predicting this number of structures would not be feasible. The strategy was to select one representative for every VFAM and cluster the representatives instead of the whole VFAMs. To select a representative, we have aligned all members of VFAM to the HMM of that VFAM and selected the highest scoring member as a candidate for which the structure would be predicted by alphafold 2. After obtaining structure predictions for all representatives, we conducted the clustering using the FoldSeek tool [32] with the default settings (commit 427df8a6b5d0ef78bee0f98cd3e6faaca18f172d, command: foldseek easy-cluster). FoldSeek was used to cluster predicted structures from AlphaFoldDB [33] and was therefore an appropriate choice for our clustering task. Functional annotations of VFOLDs are obtained in the same way as for VOGs and VFAMs. Starting from VOGDB release 225, we published all predicted protein structures and their pLDDT scores with each VOGDB release.

### 2.6. Quality Assessment of the Clustering Results

The quality of the clustering was assessed by the homogeneity of functional annotations of cluster members and the structural superfamily membership of cluster members. The homogeneity of clusters from VOGDB was compared to the homogeneity of a random model. We obtained the random model by randomly scrambling functional annotation keywords or structure superfamily labels between annotated proteins and calculated the homogeneity. The randomization step was repeated 1000 times. For functional annotations, we searched SwissProt with all protein members of a group, used the keyword of the top-level functional annotation of the hits and calculated the relative frequency of the most common annotation compared to all retrieved annotations. To assess the homogeneity of structural patterns, we used a similar approach, but instead of searching SwissProt, we searched the astral95 database (v2.08) [19] using cd-hit [34]. Hits were associated with protein structural superfamilies as described in the SCOPe database [19]. The homogeneity for superfamilies was calculated as the relative frequency of the most common superfamily per group. Comparison of the homogeneity to the random model was made using the Kolmogorov–Smirnov test.

## 3. Results

### 3.1. Database

As the RefSeq database is updated bimonthly, VOGDB is updated with every RefSeq release, and the new release is made available shortly after the newest version of RefSeq is released. The release number of VOGDB is the same as the release number of RefSeq, which was used to build it. As an example in the text, the VOGDB version 221 based on the RefSeq 221 will be used, and it contains 14,974 VOGDB genomes. The polyprotein reannotation step predicted 5499 additional peptides from 995 polyproteins.

### 3.2. Content

In the VOGDB release 221, 606,019 viral proteins were clustered and produced 59,196 VOGs, 38,576 VFAMs and 30,516 VFOLDs (Figure 1). Due to the clustering, 352,350 (58%) proteins have functional annotation compared to 333,379 (55%) of the initial proteins from RefSeq that were not annotated as hypothetical proteins. The size distribution of the groups from all three layers shows the expected pattern observed in the similar databases where there are many of the smaller groups and a few of the larger groups. The distribution of the VOGs, VFAMs and VFOLDs according to their size is visualized in Figure 2. A feature of VOGs, VFAMs and VFOLDs is the information on the lowest common ancestor (LCA) of the viruses contributing proteins to the groups. Particularly interesting groups are those with LCA “viruses”, which means that proteins from different viral realms were clustered together. There are 2441 such VOGs (4.1%), 1443 VFAMs (3.7%) and 1515 VFOLDs (4.8%). Three files containing the lists of these clusters are available online under https://vogdb.org/evalution/vogdb221.

### 3.3. Quality Assessment

As there is not yet a universal standard procedure to evaluate the clustering of the viral proteins into orthologous groups, we assessed the quality of the VOGDB clusters using the homogeneity of functional annotation and structural classification. If the clustering would perfectly group the proteins by structure and function, all proteins in one cluster would have the same and unique functional and structural annotation. The level of granularity needs to ensure maximal information for the entire database. Too coarse granularity would overestimate the homogeneity, and too fine would underestimate it. We selected the SwissProt keywords and the SCOPe superfamilies as the granularity level at which we calculate the homogeneity. Because there is a limited number of keywords describing the function, we estimated the baseline of the homogeneity from the random model described earlier. Quality assessment based on the homogeneity (Figure 3) shows that both the functional and structural homogeneity of groups from different layers of VOGDB are significantly larger than the baseline for all of the size bins (Kolmogorov–Smirnov test, *p*-value < $10^{-5}$).

### 3.4. Comparison with Similar Databases

To evaluate the homogeneity of functional annotations and structural features, we calculated the homogeneity of the COG database (the release from 2020) [35], the PHROG database (v3) [14] and the pVOG database (May 2016) [12] in the same way as for the VOGDB layers. Clusters in the pVOG database are created similarly as VOGs, and the PHROGs clusters are similar to VFAMs. However, VOGDB has a bigger scope than pVOG and PHROG by including both phages and eukaryotic viruses and therefore needs to cluster more and more diverse proteins. The COG database was included as a control, as it was creating using a similar clustering methodology and is manually curated. Figure 4 shows that the homogeneity of VOGDB layers is in the same range as the homogeneity of databases grouping prokaryotic orthologs (COG), phage orthologs (pVOG) and phage remote homologs (PHROG). The homogeneity of clusters from pVOG and VOGs is very similar, which is expected as both are created using COGSoft [22].

### 3.5. Availability

#### 3.5.1. VOGDB Webpage

VOGDB is accessible online at https://vogdb.org where it is possible to browse the clusters and see the statistics of the latest release. The webpage is updated regularly as a new version of VOGDB is calculated. The pre-computed files for the comparison of the VOGDB clusters with clusters from similar databases (see above) are available online under https://vogdb.org/evalution/vogdb221.

#### 3.5.2. VOGDB Release Files

Apart from being accessible via the webpage, we offer all of the resulting files for download. The files offered are formatted similarly to the files offered by the EggNOG database [36]. The most important files offered are HMMs of the clusters and multiple sequence alignments, files with the lowest common ancestry, files with a functional annotation of clusters, an interactive chart of genome taxonomies and predicted structures of VFAM representatives.

## 4. Discussion

### 4.1. Limitations

VOGDB is so far the most complete database for virus orthologous groups, virus protein families and virus protein structural similarities. However, it is based on the annotations provided by the underlying RefSeq database. So far, several annotation quality filters and the reannotation of polyproteins are the only means that VOGDB uses to ensure the high accuracy of its input data. A consistent reannotation of all virus genomes is not in the scope of VOGDB. Nevertheless, such reannotation will become increasingly important to sustain the value of comparative genomics of viruses. The VOGDB groups can be a valuable tool toward this aim, e.g., by predicting protein-coding genes that were missed in the original genome annotations.

### 4.2. Support for Bioinformatic Workflows

Viral hallmark genes [37] could be defined as genes that are found in diverse viruses but have no or only few homologs in cellular organisms and are therefore indicative of the viral origin of a sequence. HMMs of VOGs and VFAMs that represent viral hallmark genes can be used to predict viral sequences from unknown genomes [38] and to estimate the contamination of a viral sequence with bacterial genes [39]. HMMs of groups of viral proteins (either hallmark or not) could be used as input for various other tools. For example, the tool HMM-GraspX [40] uses protein family HMMs to guide the assembly, which is useful if the aim of the analysis is to analyze viruses in samples with a low abundance of viral reads or when the focus on specific families is needed [41]. For VOGDB clusters, we calculate the virus specificity based on the number of hits to cellular organisms based on the HMM–HMM search to the most recent eggNOG database [36]. This database contains selected representatives for cellular species and shows less study bias than genome sequence archives. We therefore approximate virus specificity by allowing for hits in maximally two, three or four cellular genomes with decreasing e-Value thresholds. The virus specificity information can be used to identify clusters representing the viral hallmark genes. Table 1 shows the number of virus-specific VOGs and VFAMs at different stringency criteria, accepting few cellular homologs as expected, e.g., from proviruses.

### 4.3. Usage for Metagenome Analysis

VOGDB is useful for analyzing metagenomic datasets that intentionally or accidentally contain virus nucleic acid sequences. When pair-wise sequence database searches fail to reveal hits, homology searches with databases containing HMMs, such as from VOGDB, are more sensitive and allow for more proteins to be annotated. In addition to the functional annotation, lineage information of the genome carrying the gene can be inferred. By mapping all genes of a viral contig to VFAMs and using the information about the lowest common ancestor of VFAMs, one can estimate the virus origin of the whole contig. The performance of profile hidden Markov model databases, including VOGDB, for virus identification has recently been evaluated across multiple application scenarios, utilizing both simulated and real metagenomic data [42].

## 5. Conclusions

VOGDB is a novel resource in the field of virus bioinformatics, and it offers unique features compared to the similar databases and will complement the current toolbox for studying viral genomes. By including both phages and eukaryotic viruses from RefSeq, VOGDB has the biggest scope of all virus orthology databases, and it still ranks similarly with them in terms of the homogeneity of functional annotations and structural classes. The three layers of grouping give the opportunity to analyze the gene clusters connected by the increasingly remote similarity. Downloadable files, including functional annotations of clusters and HMMs, as well as bi-monthly updates that follow the RefSeq releases, make VOGDB a universal tool for downstream workflows in virus bioinformatics.

VOGDB is under constant development, and new knowledge about viruses is quickly implemented (for example, the new phage taxonomy [43]). On the other hand, the stable naming of clusters allows for the comparability of the results obtained by different releases of the database. VOGDB will also be further developed with respect to the user needs and to novel computational algorithms. With release 221, we have replaced the one clustering level with three clustering levels, including structural similarity. This has improved the usabilility of VOGDB for studying viral protein families without detectable sequence similarity. In the future, we will incorporate improved methods for structure prediction as they become available. We also will improve the reproducibility of VOGDB creation by making the entire database creation workflow open source. Finally, we plan to implement typical VOGDB-driven workflows, such as virus genome annotation or the classification of metagenomic contigs, as web-based open-source pipelines. We invite the user community to share their experience with us and inform us about their needs.

## Figures and Tables

**Figure 1 viruses-16-01191-f001:**
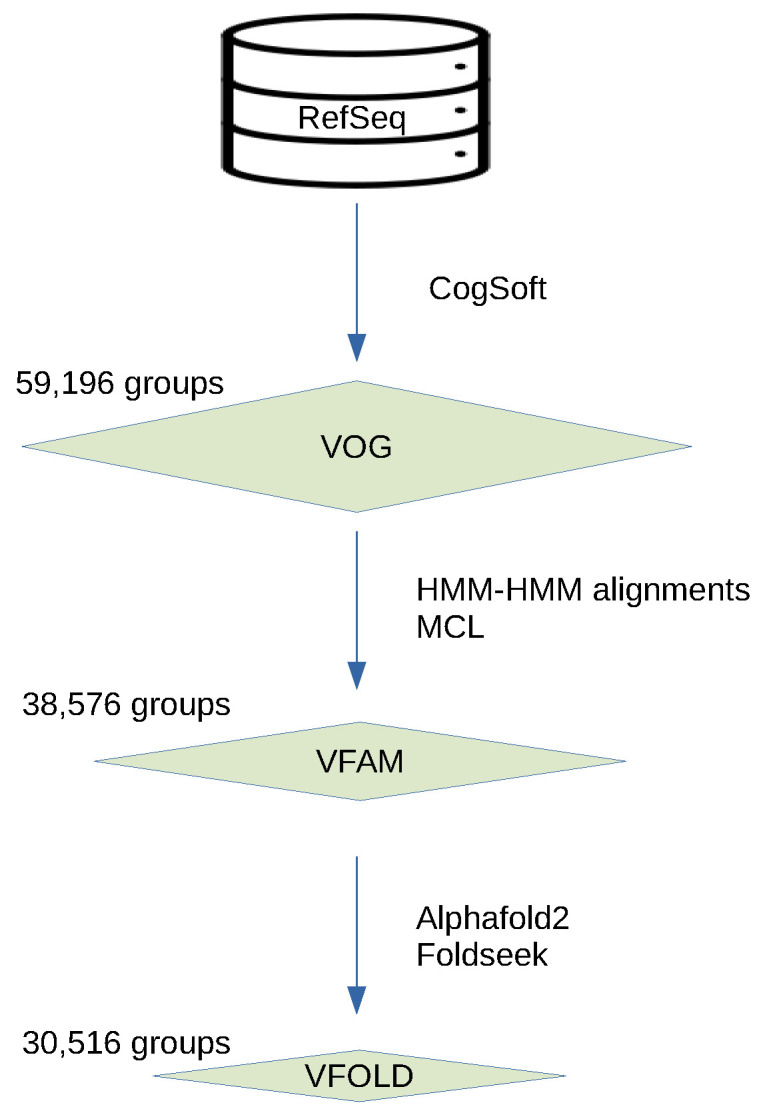
Schema of the layered structure of the database. For each layer, different tools were used to create clusters. Clusters from every next layer are built from the clusters of the previous layer and are connected by more remote similarity.

**Figure 2 viruses-16-01191-f002:**
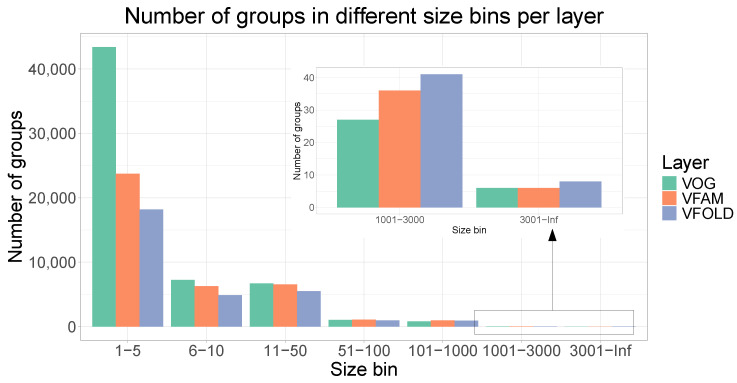
Number of groups per layer in different size bins. Size bins represent the range of the number of proteins for groups in a certain bin. The distribution with many smaller clusters and fewer of the larger ones is what is also observed in the similar databases.

**Figure 3 viruses-16-01191-f003:**
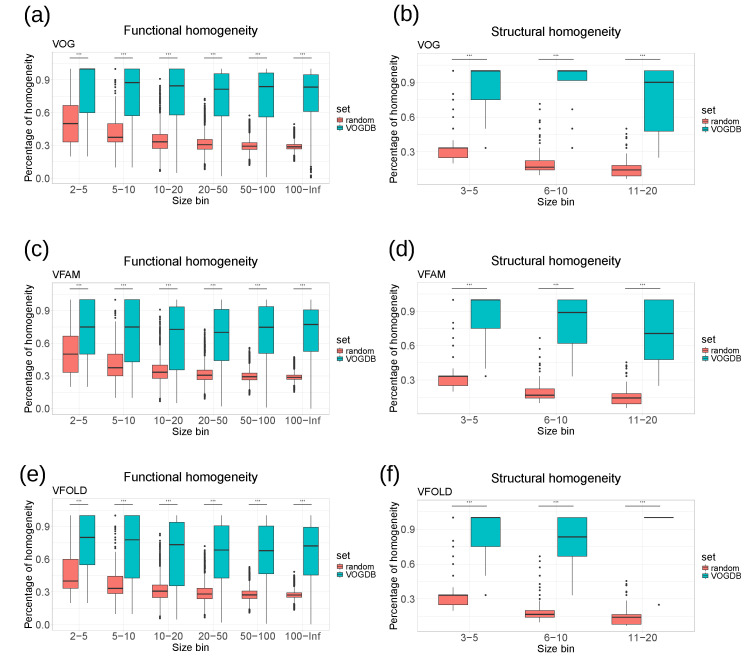
Homogeneity of functional annotations and protein structure classifications in VOGDB layers compared to the random model. (**a**–**f**) The groups from each layer are put into size bins based on the number of proteins with functional and structural annotation. The random model is created by randomly redistributing the functional and structural annotation labels between the proteins with respective annotation 1000 times and calculating the overall homogeneity. The results show that groups from VOGDB layers are significantly more homogeneous in terms of SwissProt keywords and structural classifications based on the SCOPe superfamilies (Kolmogorov–Smirnov test, *p* < $10^{-5}$).

**Figure 4 viruses-16-01191-f004:**
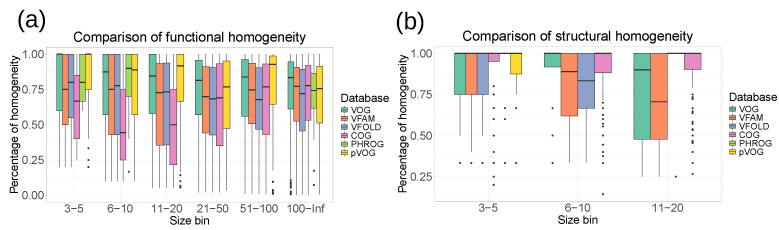
Homogeneity of SwissProt keywords (**a**) and SCOPe superfamilies (**b**) for layers from VOGDB and the other databases with orthologous/homologous groups: pVOG (phage orthologous groups), PHROG (phage remote orthologous groups) and COG (prokaryotic orthologous groups). The databases are split into size bins according to the number of proteins with a functional or structural annotation. Bins containing less than 3 proteins are not shown. The results show that the function and structure-based homogeneity of the layers from VOGDB are in the same range as in other similar databases.

**Table 1 viruses-16-01191-t001:** Virus specificity of vFAMs. Virus-specific vFAMs are useful for identifying the viral hallmark genes, the genes definitive for the viral state and with only a very remote similarity to cellular genes. In VOGDB, viral specificity is defined with three stringency levels: strict, medium and low with hits to maximally two, three or four cellular genomes with e-values up to $10^{-4}$, $10^{-10}$ and $10^{-15}$.

Layer	Strict	Medium	Low
vOG	38,562	45,613	48,627
vFAM	28,500	32,546	33,951

## Data Availability

Data available in a publicly accessible repository.

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
