# Peer review of "VOGDB—Database of Virus Orthologous Groups"

_viruses, 2024, doi:10.3390/v16081191_

Round 1

Reviewer 1 Report

Comments and Suggestions for Authors

Trgovec-Greif et al. present a new database of viral orthologous groups on three different levels. The database is created in a straightforward way, regularly updated and will be of use for the community.The manuscript should be accepted for publication.

The database seems thoroughly curated since the authors filter and partially reannotate (poly)proteins found in RefSeq. The hierarchical clustering into VOGs, VFAMs and VFOLDs is elaborate, and all are functionally annotated. A taxomonic annotation of their clusters allows for an analysis of viral contigs which can then be allocated according to the lowest common ancestor. This makes the database useful for other researchers.

I just have very minor remarks:

Section "functional annotation": What are "vVGs"? Probably a typo.

Section "Comparison with similar databases": "form" -> from

Author Response

Response to editor:

Comments 1: Three reviewers found your study interesting and VOGDB database appealing. They raised concerns and offered insightful suggestions that must be addressed. Also:

Response 1: We thank the editor for reviewing our manuscript. Your recommendations were very helpful and we have incorporated them into the manuscript.

Comments 2: - consider using remote similarities instead of remote homology, since homology is binary state

Response 2: Yes, we agree. We have modified the sentences in the manuscript, in which the remote similarity is meant quantitatively. Changes have been made in lines 8,10, 302, Table1, Figure 1.

Comments 3: - consult https://doi.org/10.1093/bioinformatics/btaa065 for impact of protein target size on estimation of hit statistical significance

Response 3: Thanks for this suggestion. We have cited this interesting paper and mentioned it for the protein target size on estimation of hit statistical significance. Changes have been made in lines 101ff.

Response to Reviewer 1:

Comments 1: Trgovec-Greif et al. present a new database of viral orthologous groups on three different levels. The database is created in a straightforward way, regularly updated and will be of use for the community.The manuscript should be accepted for publication.

The database seems thoroughly curated since the authors filter and partially reannotate (poly)proteins found in RefSeq. The hierarchical clustering into VOGs, VFAMs and VFOLDs is elaborate, and all are functionally annotated. A taxomonic annotation of their clusters allows for an analysis of viral contigs which can then be allocated according to the lowest common ancestor. This makes the database useful for other researchers.

I just have very minor remarks:

Response 1: We thank the reviewer for reviewing our manuscript. Your recommendations were very helpful and we have incorporated them into the manuscript.

Comments 2: Section "functional annotation": What are "vVGs"? Probably a typo.

Response 2: Thanks, we have corrected this typo on line 127.

Comments 3: Section "Comparison with similar databases": "form" -> from

Response 3: Thanks, we have corrected this typo on line 231.

Reviewer 2 Report

Comments and Suggestions for Authors

Reliable annotation of viral genomes is not a feasible task, requiring the use of complex methods and a certain level of knowledge of viral genomics. The manuscript by Trgovec-Greif and colleagues describes a popular and useful tool, VOGDB (Database of Virus Orthologous Groups), which can facilitate the annotation process and improve the quality of annotations. The authors explained how VOGDB was created and updated, they showed a simple logic scheme for clustering viral proteins using conventional homology search, HMM search and structural analysis. The methods used seem to be consistent and allow high-quality clustering.

The manuscript design is straightforward and the data and comparative analysis presented are compelling. The figures are attractive and comprehensive. Overall, the paper may be considered for publication in the journal Viruses. However, the manuscript could benefit from addressing some minor issues.

- “Contrary to the prokaryotic genomes from RefSeq, where PGAP [16] is used for the consistent re-annotation of genomes, virus genomes in RefSeq keep their annotation from their GenBank [17] submission.” - please check this. As far as I remember, when submitting a bacterial genome to GenBank, the submission system offers to annotate it using PGAP. Concerning phage genomes, RefSeq annotations can differ from GB annotations.

- Perhaps, the main problem when annotating a virus is the poor quality of annotations of RefSeq and NCBI viral sequences - how does VOGDB handle this problem?

- Perhaps, the last paragraph of the Introduction section should belong to the Results section.

- How would you score the quality of FoldSeek search compared to DALI, for example? Why did you choose this tool?

- Please use italics in the names of viral taxa (Fig. 5 and caption).

- How has VOGDB changed since it first appeared? Do you plan to add new features which can ease the annotations procedure? Perhaps, you should mention it in the manuscript.

Author Response

Response to Reviewer 2:

Comments 1: Reliable annotation of viral genomes is not a feasible task, requiring the use of complex methods and a certain level of knowledge of viral genomics. The manuscript by Trgovec-Greif and colleagues describes a popular and useful tool, VOGDB (Database of Virus Orthologous Groups), which can facilitate the annotation process and improve the quality of annotations. The authors explained how VOGDB was created and updated, they showed a simple logic scheme for clustering viral proteins using conventional homology search, HMM search and structural analysis. The methods used seem to be consistent and allow high-quality clustering.

The manuscript design is straightforward and the data and comparative analysis presented are compelling. The figures are attractive and comprehensive. Overall, the paper may be considered for publication in the journal Viruses. However, the manuscript could benefit from addressing some minor issues.

Response 1: We thank the reviewer for reviewing our manuscript. Your recommendations were very helpful and we have incorporated them into the manuscript. 

Comments 2:- “Contrary to the prokaryotic genomes from RefSeq, where PGAP [16] is used for the consistent re-annotation of genomes, virus genomes in RefSeq keep their annotation from their GenBank [17] submission.” - please check this. As far as I remember, when submitting a bacterial genome to GenBank, the submission system offers to annotate it using PGAP. Concerning phage genomes, RefSeq annotations can differ from GB annotations.

Response 2: Indeed, this statement needed a revision. We have clarified and improved it in the manuscript on lines 66-68.

Comments 3:- Perhaps, the main problem when annotating a virus is the poor quality of annotations of RefSeq and NCBI viral sequences - how does VOGDB handle this problem?

Response 3: We agree, this is a very important aspect. We have extended the explanation in the manuscript on lines 125ff.

Comments 4:- Perhaps, the last paragraph of the Introduction section should belong to the Results section.

Response 4: Thanks for this recommendation. We have moved most of the last paragraph of the introduction to a first subsection of the methods, introducing the general concept. Lines 69ff.

Comments 5:- How would you score the quality of FoldSeek search compared to DALI, for example? Why did you choose this tool?

Response 5: Foldseek was selected as it was already used to cluster structure predictions from the AlphafoldDB. Dali was not used since to our knowledge it is only available as a webserver. As such, it cannot be reasonably integrated into the automatic software pipeline for VOGDB creation every 2 months, while it’s performance is to our knowledge comparable to the performance of FoldSeek. We have added the AlphafoldDB citation at line 173.

Comments 6:- Please use italics in the names of viral taxa (Fig. 5 and caption).

Response 6: Following a recommendation of reviewer 3, we have removed Figure 5 entirely from the manuscript. It was replaced by a reference to a recent evaluation of HMM databases for viral metagenomics. See lines 291ff.

Comments 7:- How has VOGDB changed since it first appeared? Do you plan to add new features which can ease the annotations procedure? Perhaps, you should mention it in the manuscript.

Response 7: Thanks for this recommendation. We have extended the conclusion  in the end of the manuscript and explain the changes in VOGDB as well as future plans. Lines 309ff.

Reviewer 3 Report

Comments and Suggestions for Authors

Congratulations for this great work. Having it tied to RefSeq and updating is great, as well as having three layers of clustering including structural comparisons. The manuscript is very well written and easy to understand.

I thik this is a valuable ressource for viral bioinformatics, here are my thoughts and suggestions on the paper: 

I really like your layers of clusterings, but I think the paper would benefit from some clarifications: 

1) HMM-HMM comparisons with hh-suite were not filtered with any threshold before clustering ? That is a big concern because these type of profile-profile similarities can gather false homologs...

2) FoldSeek comparisons are also not detailed, what thresholds were used to identify significant structural similarity ? 

3) What are the coverage / alignment fraction thresholds ? Although it is part of points 1 and 2, I'm putting this in a separate point because it is key to avoid transitivity issues.

A quick way to check this is to:

1) Have as supplementary material boxplots of identity percentage and alignement fraction for each of the clusters, for the three layers, and at each bin size like on figure 2

2) Compare the clusters to external databases like PFAMs, PHROGs, pVOGs etc. 

In addition to SwissProt and astral95, it could be nice to have the results of these comparisons pre-computed and available to download, because a user will often combine several databases for cross-validation anyway.

Also this will help identify any "wrong clusters" containing false orthologs, and understanding obscure annotation terms (for a non-expert) coming from SwissProt or RefSeq.

About the structure predictions with alphafold2 what is the confidence score on these structures ? Maybe you could report this somewhere in supplementary material just to be sure that some VFOLDs were not generated based on poor quality predictions. 

Or if you used a threshold on the predictions, reporting it could also help.

Additional small thoughts:

L189 - Could you report the increase in annotation as a percentage as well ? it would be helpful for readers.

With that I think it's important to state how many proteins are singletons and their percentage on the total dataset (alone in a cluster).

L197 - What annotations are in these VOGs that have "Viruses" as their LCA ? And are these consistent with the literature ? Maybe a supplementary figure could be helpful here.

L220 - "a harder clustering task" I think that is a broad but powerful statement, it requires some explanation, maybe supported by a reference. Why is it a harder task when adding eukaryotic viruses ? 

A useful figure could be a pie chart of the taxonomic diversity across all genomes used here, this is not necessary but very informative. Maybe this exists somewhere in NCBI's website since it's RefSeq but I am not aware. If some taxonomic groups dominate VOGDB then implicitly these will be the better annotated on a user's dataset, and that's good to know beforehand.

Table 1 - How do you define these categories ?

How do you know if the two/three/four cellular genomes having homologs contain prophages or not ?

Figure 5 and the paragraph starting L263 are very "out there" I think these are "dangerous" statements and this will not be the primary use of VOGDB. Taxonomy of viruses is a sensitive and complex subject, I would advocate for the removal this section and this figure, it is not a crucial part of the paper and removing it will not affect the quality of the paper, which is great.

Author Response

Response to Reviewer 3:

Comments 1: Congratulations for this great work. Having it tied to RefSeq and updating is great, as well as having three layers of clustering including structural comparisons. The manuscript is very well written and easy to understand.

 I thik this is a valuable ressource for viral bioinformatics, here are my thoughts and suggestions on the paper: 

 I really like your layers of clusterings, but I think the paper would benefit from some clarifications: 

Response 1: We thank the reviewer for reviewing our manuscript. Your recommendations were very helpful and we have incorporated them into the manuscript.

Comments 2: 1) HMM-HMM comparisons with hh-suite were not filtered with any threshold before clustering ? That is a big concern because these type of profile-profile similarities can gather false homologs...

Response 2: Thanks, good point. Of course, the HMM-HMM comparisons are filtered. We use three different filtering criteria, which all need to be passed. We have added this information on lines 152ff.

Comments 3: 2) FoldSeek comparisons are also not detailed, what thresholds were used to identify significant structural similarity ? 

Response 3: Thanks for this recommendation. We have added a reference to the FoldSeek version (commit hash) to the manuscript as well as the command used for clustering. The default settings of the Foldseek cluster process are available on the Foldseek github page. See line 171ff.

Comments 4: 3) What are the coverage / alignment fraction thresholds ? Although it is part of points 1 and 2, I'm putting this in a separate point because it is key to avoid transitivity issues.

 A quick way to check this is to:

 1) Have as supplementary material boxplots of identity percentage and alignement fraction for each of the clusters, for the three layers, and at each bin size like on figure 2

 2) Compare the clusters to external databases like PFAMs, PHROGs, pVOGs etc. 

Response 4: Thanks for this interesting suggestion. Indeed, the quality of the multiple sequence alignments is crucial, as these are the basis for the entire downstream process. We comply with the minimum reporting standard for multiple sequence alignments https://academic.oup.com/nargab/article/2/2/lqaa024/5819602?login=true. Following your request, we have added the alistat scores for all multiple sequence alignments to the online data repository supplementing this manuscript. The repository also contains the comparative boxplots that you have suggested. Starting from VOGDB release 225, we will also provide the alistat scores for each VOGDB release with the release files. We have added the method reference in line 117 and 157.

Comments 5: In addition to SwissProt and astral95, it could be nice to have the results of these comparisons pre-computed and available to download, because a user will often combine several databases for cross-validation anyway.

Also this will help identify any "wrong clusters" containing false orthologs, and understanding obscure annotation terms (for a non-expert) coming from SwissProt or RefSeq.

Response 5: We agree that it would be valuable to share these pre-computed files, to compare the VOGDB clusters with clusters from similar databases. We added these files to the VOGDB data repository, and created a specific page for these files on the VOGDB webpage. All data are available for download now. A README file with descriptions is included as well. A link has been added at line 246.

Comments 6: About the structure predictions with alphafold2 what is the confidence score on these structures ? Maybe you could report this somewhere in supplementary material just to be sure that some VFOLDs were not generated based on poor quality predictions. 

Or if you used a threshold on the predictions, reporting it could also help.

Response 6: Indeed, thanks for bringing this up. It is a very important topic, which we are currently working on. So far, we didn't use confidence scores to filter Alphafold predictions. We are aware of the pLDDT scores. Several authors have used the mean pLDDT scores to evaluate predictions. However, so far there is no literature about mean pLDDT scores of viral protein structure predictions, on which we would base our cutoff threshold. As further literature becomes available, we plan to implement the filtering based on the mean pLDDT or another confidence score, which might be introduced with new methods such as Alphafold 3 and follow up implementations.

To address your recommendation immediately, we will start publishing all predicted structures, including their pLDDT scores, along with each VOGDB release. This will allow users to assess the predicted structures and their confidences directly. The protein structures will already be available in VOGDB release 225, which is currently being calculated. A statement has been added at line 175.

Comments 7: Additional small thoughts: 

L189 - Could you report the increase in annotation as a percentage as well ? it would be helpful for readers. With that I think it's important to state how many proteins are singletons and their percentage on the total dataset (alone in a cluster).

Response 7: Thanks for this recommendation. We have added the percentages to the numbers of annotated proteins in line 204.

Comments 8: L197 - What annotations are in these VOGs that have "Viruses" as their LCA ? And are these consistent with the literature ? Maybe a supplementary figure could be helpful here.

Response 8: Yes, we agree. We have added three respective lists to the online data repository supplementing this manuscript. We have added the reference in line 213.

Comments 9: L220 - "a harder clustering task" I think that is a broad but powerful statement, it requires some explanation, maybe supported by a reference. Why is it a harder task when adding eukaryotic viruses ? 

Response 9: Indeed, this phrase needed clarification. We have improved this statement in line 235.

Comments 10: A useful figure could be a pie chart of the taxonomic diversity across all genomes used here, this is not necessary but very informative. Maybe this exists somewhere in NCBI's website since it's RefSeq but I am not aware. If some taxonomic groups dominate VOGDB then implicitly these will be the better annotated on a user's dataset, and that's good to know beforehand.

Response 10: Thanks, this is a good point. We are not aware of such figures for RefSeq on the NCBI pages. We think the best way to explore the taxonomy of VOGDB is an interactive Krona chart. We have added it to the online data repository supplementing this manuscript and also plan to add such a figure to the upcoming release 225 of VOGDB and any future release. We have extended the statement in line 246ff.

Comments 11: Table 1 - How do you define these categories ?

How do you know if the two/three/four cellular genomes having homologs contain prophages or not ?

Response 11: These numbers were derived based on the philosophy of EGGNOG, which is used as a reference database for cellular genomes. EGGNOG provides a selection of cellular genomes, which is much less study biased than genome sequence archives. Especially for bacterial species, only few selected strains are included in EGGNOG. Assuming species specificity of prophages, they can be expected to only occur in one EGGNOG genome. So we choose the values two/three/four for the virus specificity evaluation. We have extended the paragraph in lines 277ff.

Comments 12: Figure 5 and the paragraph starting L263 are very "out there" I think these are "dangerous" statements and this will not be the primary use of VOGDB. Taxonomy of viruses is a sensitive and complex subject, I would advocate for the removal this section and this figure, it is not a crucial part of the paper and removing it will not affect the quality of the paper, which is great.

Response 12: We thank you for this critical assessment. We understand it. Fortunately, a recent evaluation review has analyzed the use of HMM based databases for virus bioinformatics, and includes VOGDB. We have therefore shortened this section, removed figure 5 and replaced it by a reference to https://academic.oup.com/bib/article/25/4/bbae292/7696516. Changes have been made in lines 291ff.